# Association between Cerebrospinal Fluid and Serum Biomarker Levels and Diagnosis, Injury Severity, and Short-Term Outcomes in Patients with Acute Traumatic Spinal Cord Injury

**DOI:** 10.3390/diagnostics13101814

**Published:** 2023-05-22

**Authors:** Zhihui Yang, Seza Apiliogullari, Yueqiang Fu, Ayah Istanbouli, Sehajpreet Kaur, Iktej Singh Jabbal, Ahmed Moghieb, Zoha Irfan, Robert Logan Patterson, Milin Kurup, Lindsey Morrow, Michael Cohn, Zhiqun Zhang, Jiepei Zhu, Ronald L. Hayes, Helen M. Bramlett, M. Ross Bullock, W. Dalton Dietrich, Michael Y. Wang, Firas Kobeissy, Kevin W. Wang

**Affiliations:** 1Program for Neurotrauma, Neuroproteomics & Biomarkers Research, Department of Emergency Medicine, University of Florida, Gainesville, FL 32611, USA; 2Department of Neurobiology, Center for Neurotrauma, Multiomics & Biomarkers (CNMB), Neuroscience Institute, Morehouse School of Medicine, 720 Westview Dr SW, Atlanta, GA 30310, USA; 3Banyan Biomarkers, Inc., 16470 West Bernardo Drive, Suite 100, San Diego, CA 92127, USA; 4Department of Neurological Surgery, Leonard M. Miller School of Medicine, University of Miami, Miami, FL 33136, USA; 5Brain Rehabilitation Research Center, Malcom Randall Veterans Affairs Medical Center (VAMC), 1601, The Archer Rd., Gainesville, FL 32608, USA

**Keywords:** spinal cord injury, ASIA impairment scale, American Spinal Injury Association, biomarker, injury severity, diagnostic and prognostic markers, αII-spectrin breakdown product, ubiquitin C-terminal hydrolase-L1, S100 calcium-binding protein-B, glial fibrillary acidic protein, GFAP breakdown product, neurofilament light chain, interleukin

## Abstract

Acute traumatic spinal cord injury (SCI) is recognized as a global problem that can lead to a range of acute and secondary complications impacting morbidity and mortality. There is still a lack of reliable diagnostic and prognostic biomarkers in patients with SCI that could help guide clinical care and identify novel therapeutic targets for future drug discovery. The aim of this prospective controlled study was to determine the cerebral spinal fluid (CSF) and serum profiles of 10 biomarkers as indicators of SCI diagnosis, severity, and prognosis to aid in assessing appropriate treatment modalities. CSF and serum samples of 15 SCI and ten healthy participants were included in the study. The neurological assessments were scored on admission and at discharge from the hospital using the American Spinal Injury Association Impairment Score (AIS) grades. The CSF and serum concentrations of SBDP150, S100B, GFAP, NF-L, UCHL-1, Tau, and IL-6 were significantly higher in SCI patients when compared with the control group. The CSF GBDP 38/44K, UCHL-L1, S100B, GFAP, and Tau levels were significantly higher in the AIS A patients. This study demonstrated a strong correlation between biomarker levels in the diagnosis and injury severity of SCI but no association with short-term outcomes. Future prospective controlled studies need to be done to support the results of this study.

## 1. Introduction

Acute traumatic spinal cord injury (SCI) may result in permanent damage to the spinal cord, and can have devastating physical, psychological, and economic consequences for patients, their families, and society [1]. In the US, there is an annual incidence of SCI of approximately 18,000 new cases per year [2]. While the chances of SCI occurring in individuals is relatively low (around 54 patients per one million in the USA [2,3,4], young adults and the elderly are mostly affected, with the average age climbing from 29 years during the 1970s to 43 since 2015 [1]. The majority of SCIs are due to road traffic crashes and falls [1], always resulting in neuronal injury [5]. Most of the patients with SCI suffer from a severe sensory deficit and neuropathic pain and experience multisystem dysfunctions, including the urinary, gastrointestinal, and pulmonary systems and autonomic dysfunction [6]. In aged patients, minor trauma caused by falls and other accidents can sometimes result in SCI [7]. In a study that evaluated injury-related mortality and disability in six European countries, SCI was found to be the second leading cause of morbidity and disability after traumatic brain injury (TBI) [8].

In patients with suspected spinal trauma, a correct diagnosis should be made in a timely manner to reduce the occurrence of secondary medical issues that could worsen long-term outcomes. Timely neuroprotective interventions including early surgery following the rapid localization and classification of neurological injuries can have critical benefits and affect the long-term neurological recovery of patients [9,10,11]. For this reason, current international guidelines suggest reducing the time from injury to surgery, and support the concept of “time is spine” [10]. On the other hand, missing the medical diagnosis of SCI can have dramatic consequences, as can the medico-legal implications associated with them [12]. It is important to note that established clinical evaluations that are commonly used may not apply to all patients exposed to traffic accidents, and may be difficult to use in patients with undefined traumas [7]. For example, in the absence of radiological findings using computed tomography of the spine, a diagnosis of SCI may be easily overlooked [12]. In this regard, the identification of surrogate biomarkers through rapid diagnostic tests holds promise for the earlier diagnosis of injury severity, decreases the therapeutic window for surgical interventions, and helps identify patients that may require specialized interventions.

The pathophysiology of acute traumatic SCI is complex and can be divided into primary injuries and secondary injuries [13,14]. In the primary phase, immediate acute cell dysfunction and cell death occur, which are often associated with vertebral fracture, dislocation, hematoma, ligament tear, and soft tissue damage [13,15]. After the primary injury event, a cascade of secondary injury mechanisms would follow. This often involves vascular, cellular, and biochemical events, further aggravating the zone of neural tissue injury and exacerbating neurological deficits [9,13,15]. Secondary injury can set in within minutes and can last for weeks or even months [13,14]. Primary injury occurs suddenly and is associated with delays of between several hours to days before specialized care is administered [10,13,15]. Currently, innovative research is being conducted to recognize novel secondary injury mechanisms with the goal of identifying new therapeutic targets for neuroprotection and repair [13,14,16].

Developing strategies to accurately determine the degree of injury and to predict neurological outcomes following SCI is another important issue for successfully treating acutely injured patients. However, this is a complex task considering the heterogeneous nature of the SCI population, including both the nature and location of the primary injury, the subsequent secondary injury mechanisms, as well as the lack of effective pharmacological treatments to promote neurological recovery [13,17]. Currently, The American Spinal Injury Association Impairment Score (AIS) is universally used to classify SCI severity and is often considered the best prognostic indicator. The severity of the primary injury is often used against the AIS benchmark [13,14]. However, it is also appreciated that the exacerbating effects of secondary injury processes can provoke more damage; therefore, the assessment of these processes in the individual patient may be a good predictor of a patient’s long-lasting neurological consequences [13,14]. The identification and critical measurement of these surrogate markers, including advanced imaging strategies as well as biofluid-based biomarkers reflecting the severity of neurological injury, would have great potential as predictive prognostic markers of SCI [18].

Elevated concentrations of biomarkers have been observed in both the serum and CSF after TBI and SCI [16,19]. In addition, a CSF analysis has the potential to identify blood biomarkers for the diagnosis, SCI severity, and prognosis. Although it, therefore, seems reasonable to investigate the changes in CSF markers, this approach is difficult to assess, since CSF is not routinely collected and is not considered part of the standard management of SCI [20]. In addition, a lumbar puncture is an invasive procedure [21] and it may also produce serious complications, including subdural hematoma [22]. For these reasons, blood or other biofluid-based non-invasive tests including urine or saliva are the standard of care. Based on the current biomarker information for clinical SCI, there is a great need for the rapid and reliable detection of injury-sensitive SCI biomarkers.

To date, biomarkers of SCI have been identified from mostly CSF and animal studies [23]. Amongst the clinical studies, there is a limited number of articles that have assessed CSF or serum samples in patients with SCI [16,18,20,24,25]. Several candidate biomarkers and their breakdown products have been investigated in the realm of traumatic brain injury (TBI) and SCI, and have been identified and validated through omics approaches applied to experimental and clinical studies [26,27,28,29,30,31,32]; among these are αII-spectrin, ubiquitin C-terminal hydrolase-L1 (UCH-L1), S100 calcium-binding protein-B (S100B), Tau, glial fibrillary acidic protein (GFAP), and neurofilament light chain (NF-L) [33,34,35,36,37,38]. In the present study, the potential protein biomarkers were evaluated from the CSF and serum samples in SCI patients as compared with controls. These biomarkers include the following sections.

The αII-spectrin protein forms part of the axolemma cytoskeleton. The 280-kDa αII-spectrin molecule is cleaved into the αII-spectrin breakdown products SBDP 145-kDa and SBDP 150-kDa [39]. SBDPs can be used to detect apoptosis [33,34]. In previous studies, significant elevations of SBDP have been reported both in CSF and serum samples during the first 24 h following experimental SCI [23] and in patients with TBI [34]. Although SBDPs may be important biomarkers to assess in SCI patients, to the best of our knowledge, there is only one published preliminary research article that has evaluated SBDPs as biomarkers for patients with SCI [16]. Ubiquitin C-terminal hydrolase-L1 (UCH-L1) is an important enzyme in regulating the brain protein metabolism and has been suggested to reflect degrees of neuronal cell injury after SCI [35,36,37]. Yang et al. [23] reported UCH-L1 levels in both CSF and serum to increase 4 h after experimental SCI. Because of its high expression in neurons, there have been notable clinical studies on UCH-L1 in SCI [36,37,40]. Indeed, Stukas et al. [37] showed that CSF UCH-L1 levels in SCI may be a helpful biomarker to predict SCI severity and functional outcomes. The S100 calcium-binding protein-B (S100B) is an astroglial 11 kDa calcium-binding protein found in glial cells that can also reflect the degree of SCI injury, making it a potential biomarker for the prognosis in SCI patients [18,41,42,43]. In this regard, Lee et al. [43] showed that this raised serum S100B levels in all patients with acute SCI. Tau is a microtubule-associated protein that contributes to the formation of neuronal cells. This protein has an important role in the stabilization of the internal cytoarchitecture and the transport of nutrients [25]. Clinical studies suggest that CSF Tau levels showed significant increases in patients with acute traumatic SCI [18,41,44,45]. For this reason, Tau may be considered a potential diagnostic and prognostic biomarker for SCI [25]. However, to date, no study has assessed serum Tau concentrations in patients with SCI. GFAP is another biomarker, which is a skeletal intermediate filament protein in astrocytes. Increased levels reflect the pathological state of astrogliosis and astrocyte injury [18,25,46]. GFAP is widely accepted as a diagnostic TBI biomarker [47]. A limited number of studies have shown elevated GFAP levels and GBDP 38/44K in the CSF and serum in patients with SCI [16,24,41,46,48,49].

NF-L, present in the axonal cytoplasm, marks neurodegeneration, and its presence is often used to evaluate neurological disorders. NF-L is increased in the extracellular environment following neuronal cell damage; therefore, it may be considered a potential biomarker for SCI [24,25,46,50,51]. Why NF-L concentrations between the plasma and CSF vary by between 40% to 60% remains unexplained [51]. Finally, interleukin (IL)-6, an inflammatory cytokine, has also been reported to demonstrate positive associations with injury severity and outcomes following SCI [18,41].

Based on the current biomarker information for clinical SCI, there is a great need for the rapid and reliable detection of injury-sensitive SCI biomarkers. The primary aim of this prospective controlled study was to determine the CSF and serum profiles of potential biomarkers for the diagnosis of SCI severity. The secondary aim was to determine whether these markers predict the neurological outcome with respect to AIS grade improvements at discharge from the hospital.

## 2. Materials and Methods

### 2.1. Clinical Trial Enrollment

This study underwent a review and approval process by the Institutional Review Board of the University of Miami–Jackson Memorial Hospital Healthcare System (IRB#20090655). In all cases where CSF and serum specimens were collected, informed consent was obtained as per the protocol. Patients included in the study had a known traumatic durotomy identified at the time of emergent or urgent spinal surgery requiring CSF diversion to allow healing. Thus, all intrathecal (lumbar drain) catheters were inserted for clinically necessary patient care. The inclusion criteria for acute SCI patients in this study included an injury of AIS grade A (motor and sensory complete paralysis), B (motor complete, sensory incomplete paralysis), C (incomplete motor and sensory paralysis), or D (sensory incomplete paralysis) upon presentation. Patients aged ≥ 18 years with both cervical and thoracic spinal injuries were included, and a neurological examination was conducted within the first 48 h postinjury. Patients were excluded if there was a concomitant TBI or major trauma of any part of the body that required major surgical intervention. Other exclusion criteria included patients requiring cardiopulmonary resuscitation before enrollment; pregnant women; individuals with a known previous history of SCI or TBI; individuals who were intoxicated upon admission; individuals who had a neurological, neuromuscular, autoimmune, severe hepatic, or renal disease; and patients with malignancies or who had other contraindications for surgery.

### 2.2. CSF and Serum Collection and Processing

Lumbar drain catheters were inserted distal to L2 using a strict aseptic technique either at the time of the primary surgical intervention or shortly thereafter in the intensive care unit. All lumbar drains were inserted within 48 h of SCI. Serial CSF and blood samples were then collected at the same time as each CSF sample was removed. All efforts were made to obtain the most recently drained CSF within the catheter, and the CSF was not collected from the repository chamber to allow temporally relevant sampling. The CSF and serum samples were centrifuged at 1000 rcf for 10 min, and the supernatant was then dispersed into 200 µL aliquots and immediately frozen in an ethanol–dry ice bath and stored at −80 °C freezer as de-identified samples until used for biomarkers assays. Control CSF (*n* = 10, average age 38, male 70%) and serum samples (*n* = 10, average age 43, male 70%) were purchased from BioIVT (previously Bioreclamation Inc., Westbury, NY, USA). They were used for comparison with stored de-identified SCI CSF and serum samples.

### 2.3. Biochemical Analysis

The following ten biomarkers were assessed in the study: αII-spectrin, SBDP150, SBDP145, UCH-L, S100B, Tau, GFAP, GBDP 38/44K, NF-L, and IL-6. The biomarkers were determined using quantitative immunoblotting or an enzyme-linked immunosorbent assay (ELISA). For all analyses, we first assessed the above biomarkers individually at each time point. The first outcome assessment was between biomarker levels between the SCI group and control group levels. The second outcome was the accuracy with which biomarkers were distinguished among baseline injury severities of the AIS grades. Lastly, we also determined the association of CSF biomarker levels with AIS improvement by grade (ΔAIS = initial AIS − discharge AIS) and compared biomarker levels in patients with no Improvement from admission to discharge (ΔAIS = 0) to those with at least one grade improvement (e.g., from A to B ΔAIS = 1; from B to C ΔAIS = 1; from B to D ΔAIS = 2) to predict neurological recovery.

### 2.4. Quantitative Immunoblotting

First, 10 µL control and SCI CSF samples were mixed with 8 µM of SDS sample buffer (50 mM Tris, pH 6.8, 25 mM DTT, 2.5% SDS, 0.02% bromophenol blue, and 25% glycerol). Equal amounts of proteins were loaded onto Tris/glycine gels (Invitrogen Life Technologies, Kiryat, Israel) and then separated by electrophoresis at 200 V for 60 min. The proteins were transferred to a polyvinylidene difluoride (PVDF) membrane (Invitrogen Life Technologies, Kiryat, Israel) using the iBlot Gel Transfer Device (Invitrogen Life Technologies, Kiryat, Israel) for 7 min. Following the transfer, the membranes were blocked in 5% non-fat dry milk in TBST (20 mM Tris-HCl, 150 mM NaCl, and 0.003% Tween-20, pH 7.5) for one hour. Monoclonal anti-mouse 10 µL control and SCI CSF samples were mixed with 8 µM of SDS sample buffer (50 mM Tris, pH 6.8, 25 mM DTT, 2.5% SDS, 0.02% bromophenol blue, and 25% glycerol). Equal amounts of proteins were loaded onto Tris/glycine gels (Invitrogen Life Technologies) and then separated by electrophoresis at 200 V for 60 min. The proteins were transferred to a polyvinylidene difluoride (PVDF) membrane (Invitrogen) using the iBlot Gel Transfer Device (Invitrogen Life Technologies, Kiryat, Israel) for 7 min. Following the transfer, the membranes were blocked in 5% non-fat dry milk in TBST (20 mM Tris-HCl, 150 mM NaCl, and 0.003% Tween-20, pH 7.5) for an hour. Monoclonal anti-mouse αII-spectrin (Enzo Life Sciences, NY, USA), anti-spectrin antibody, polyclonal anti-rabbit GFAP (Abcam, Walham, MA, USA) (capable of detecting major GFAP breakdown products GBDP 38/44K), and SDBP 150/145 were incubated with immunoblotting membranes at a dilution of 1:1000 in 5% milk at 4 °C overnight. On the following day, the membranes were washed three times with Tris-buffered saline with 0.1% Tween 20 (TBST) and probed with an alkaline phosphatase conjugate goat secondary antibody (EMD Millipore, Burlington, MA, USA) at a dilution of 1:5000 in 5% milk for an hour, followed by TBST washing. Immunoreactivity was detected using 5-bromo-4-chloro-3-indolyl phosphate (BCIP)/nitro blue tetrazolium phosphatase substrate (Kirkegaard and Perry Laboratories, Gaithersburg, MD, USA). The band intensity was quantified using NIH ImageJ v1.7 software.

### 2.5. Enzyme-Linked Immunosorbent Assay (ELISA)

All assays were conducted according to the manufacturer’s protocols. The following commercial ELISA kits were used: SBDP150 (cat# MBS7606326, MyBioSource, Inc., San Diego, CA, USA, S100B (cat# EZHs100b-33K, EMD Millipore, Burlington, MA, USA) and IL-6 (cat# R6000B, R&D System, Minneapolis, MN, USA). Briefly, standards and 5–10 μL CSF or 30–50 μL serum samples were added into specific antibody precoated plates. Then, biotin-conjugated antibodies followed by HRP-conjugated avidin were incubated. A signal was detected with a chromogenic substrate (3-3-5-5-tetramethylbenzidine), and the intensity was measured using a colorimetric plate reader (450 nm).

### 2.6. Quanterix Single Molecule Array (SIMOA) N4PB Digital ELISA

The GFAP, UCH-L1, NFL, and Tau concentrations were measured using the same batch of reagents by investigators blinded to the clinical data using a Quanterix Single Molecule Array (SIMOA) Neurology 4 Plex kit in an SR-X immunoassay analyzer (Quanterix Corp, Boston, MA, USA), which runs ultrasensitive paramagnetic bead-based enzyme-linked immunosorbent assays. The Quanterix Simoa N4PB kit measures GFAP, UCH-L1, T-Tau, and NF-L serum concentrations on a multiplex array simultaneously, according to the manufacturer’s instructions (https://www.quanterix.com/products-technology/assays/neuro-4-plex-b (accessed on 1 March 2023)). The assay methodology is described in detail elsewhere and is similar to the Quanterix N4PA assay [52].

### 2.7. Neurological Evaluation

The severity of neurological impairment was graded according to the AIS scale. Baseline neurological assessments were performed upon admission (within the first 48 h postinjury). Follow-up examinations were conducted at the time of discharge from the hospital (<three months). All baseline and discharge ASI grades (A, B, C, or D) were recorded by qualified neurosurgeons or neurosurgeons-in-training. AIS improvements were graded as ΔAIS = initial AIS − discharge AIS; ΔAIS 0 = no changes; or ΔAIS 1–2 = improvements in AIS grade 1 or 2.

### 2.8. Statistical Analysis

The data were analyzed using Graph Pad Prism 7.0 (GraphPad Software, San Diego, CA, USA) and IBM SPSS statistics version 23 (IBM Corp., New York City, NY, USA). Continuous variables are described by their median and IQR, and categorical variables by numbers and percentages. Mann–Whitney U and Kruskal–Wallis tests were conducted to assess differences between groups for continuous variables. The receiver operating characteristic (ROC) was used to evaluate the diagnostic accuracy of CSF and serum biomarkers on day one between SCI and healthy control patients. All tests were two-tailed, repeated measures complemented by Bonferroni tests, and the *p*-values were adjusted for multiple comparisons. A *p*-value of <0.05 was significant, where ** *p* < 0.05, *** *p* < 0.005.

## 3. Results

A total of 15 patients with acute traumatic SCI completed the study, with 13 of 15 being males. The patient characteristics are presented in Table 1. Nine patients on admission had an AIS grade of A and six had a grade of B. Four patients showed an improvement of these grades upon discharge from the hospital. In the healthy control group, the mean ages were 33.7 ± 12.4 years, with 8 of 10 being male (CSF samples), and 35.3 ± 9.8 years, with 8 of 10 being male (serum samples).

### 3.1. CSF Biomarkers Measured with Immunoblotting

The results for biomarker (αII-spectrin, SBDP150/145, GFAP, GBDP 38/44K) levels obtained by immunoblotting in CSF and comparison between SCI and healthy control groups are presented in Figure 1. The levels of the αII-spectrin breakdown product SBDP 150/145 were significantly higher from day 1 to day 4. The GFAP and GBDP 38/44K levels were significantly higher from day 1 and day 2 in patients with SCI groups.

Figure 2 shows the ability of CSF levels of αII-spectrin, SBDP150/145, GFAP, and GBDP 38/44K to predict injury severity (AIS grade) measured by immunoblotting. The CSF levels of GBDP 38/44K on days 1 and 2 were significantly higher in patients with initial AIS grade A vs. patients with initial AIS grade B. 

The ability levels of CSF biomarker levels measured by immunoblotting to predict AIS improvements (ΔAIS = discharge AIS − initial AIS) are presented in Figure 3. There was a trend for higher biomarker levels in the no improvement group (ΔAIS = 0) when compared with the patients with grade 1 or 2 improvement (ΔAIS = 1–2) for initial CSF αII spectrin, SBDP150/145, GFAP, and GBDP 38/44K levels on day 1 to day 5. However, these differences were not statistically significant.

### 3.2. CSF Biomarkers Measured with ELISA

The SBDP150, UCH-L, S100B, Tau, GFAP, NF-L, and IL-6 levels in the CSF were assayed using an ELISA. The ability of CSF biomarkers levels in the diagnosis of SCI and a comparison between SCI and healthy control groups are presented in Figure 4.

The level of SBDP150 peaked on day 1 and was higher when compared with the control group over the 5 days. UCH-L1 showed the highest initial increase on day 1 and was still significantly high on day 3 when compared to the control levels. The GFAP and S100B levels were highest on day 1, remaining elevated up to day 5 (*p* < 0.005). Tau peaked on days 1 and 2 and gradually decreased on the subsequent days. NF-L significantly increased on day 1 and maintained its levels before peaking on day 5 (*p* < 0.005). Like the expression of NF-L, IL-6 increased on day 1 and peaked on day 5, although not as much as NF-L did (Figure 4). All biomarkers were significantly elevated on day 1 when compared to healthy controls.

Receiver operating characteristic (ROC) curves were used to assess the diagnostic value of biomarker levels based on the area under the curve (AUC; ranging from 0.7846 to 0.9769) for distinguishing healthy controls from day 1 SCI as shown in Appendix A. 

The ability of CSF SBDP150, UCH-L, S100B, Tau, GFAP, NF-L, and IL-6 levels to predict injury severity (AIS grade) measured by ELISA is presented in Figure 5.

For SCI patients with initial AIS grade A, the concentrations of SBDP150 on days 3 and 4, UCH-1 on day 2, S100B on days 2, 3, and 4, GFAP on days 2, 3 and 5, Tau on days 2, 3 and 5 were significantly higher when compared with patients with AIS grade B. However, the NF-L, and IL-6 levels were not significantly different between the AIS grade A and AIS grade B groups.

Figure 6 shows the predictive prognostic accuracy of the CSF levels of biomarkers assessed by ELISA. There were no statistically significant differences between the CSF levels of any biomarker on any day between patients with and without AIS improvement.

### 3.3. Serum Biomarkers Measured with ELISA

Figure 7 shows a comparison of the serum levels of SBDP150, UCH-L1, GFAP, NFL, Tau, and IL-6, which were statistically significantly higher in SCI patients when compared to the healthy control group from day 1 to day 5 when measured with an ELISA. Regarding the S100B concentrations, only the serum day 1 level was statistically significantly higher in SCI patients when compared to the healthy control group.

The serum SBDP150, IL-6, and GFAP levels were highest on day 1 and were still significantly higher from day 2 to day 5. The serum UCHL-1 levels peaked on day 1 and then decreased from day 2 to 5. The serum Tau levels were significantly elevated on days 1 and 2 and declined from days 3 to 5. The serum S100B levels were highest only on day 1, then decreased from day 2 to day 5. On the other hand, the serum NF-L levels were elevated on day 1 and reached the maximum on day 5. Receiver operating characteristic (ROC) curves were used to assess the diagnostic values of serum biomarkers on day 1 in patients with SCI. Area under the curve (AUC) values ranging from 0.81 to 1.00 were used to distinguish healthy controls from SCI day 1 samples (Appendix A).

Figure 8 show that ability of serum SBDP150, UCH-L, S100B, Tau, GFAP, NF-L, and IL-6 levels to predict injury severity measured by ELISA. When comparing patients of AIS grade A and AIS grade B, the serum biomarker levels are generally higher in grade A patients but only Tau and UCH-L levels were statistically significantly different. 

Compared to patients with AIS improvement and patients with no ASI improvement at discharge, none of the biomarkers showed statistically significant changes except Tau on day 2 (Figure 9).

## 4. Discussion

Although important preclinical and clinical research has been conducted to identify and test diagnostic and prognostic biomarkers in patients with SCI, encouraging findings have yet to be successfully translated into an agreed standard of care treatment for acute SCI. However, the evaluation of CSF and serum proteins in patients with SCI holds promise for subsequent studies.

The results of the present study demonstrated strong positive correlations between concentrations of biomarkers in both the CSF and serum (SBDP150, S100B, GFAP, NF-L, UCHL-1, Tau, IL-6) (Appendix A) and the diagnosis of SCI. 

We also reported that regarding the severity of injury, AIS grade A patients had significantly higher SDP 150/145, GBDP 38/44K (Figure 2), UCHL-L1, S100B, GFAP, and Tau levels in the CSF when compared with patients who had an AIS grade of B (Figure 5). Finally, regarding their predictive utility for short-term outcomes, all biomarkers had lower concentrations in patients that had a better functional short-term outcome than in patients without functional improvements. Taken together, our results suggest that the serum and CSF biomarkers evaluated in this study may have value in diagnosis, predicting injury severity and the initial recovery, and may serve as a baseline for future work. 

The 280 kDa αII-spectrin is a neuronal membrane–skeletal protein. The calpain-mediated proteolysis of αII-spectrin results in two unique and highly stable αII-spectrin breakdown products, SBDP 145-kDa and SBDP 150-kDa [39]. The high activity of calpain and SBDPs may lead to pain [39] and spasticity, which is observed in 80% of patients with SCI [53]. In this regard, animal and human studies suggest that the increased excitability of interneurons plays a major role in the pathophysiology of spasticity [39,53,54]. This excitation strongly relates to the upregulation of the persistent sodium current (INaP) and downregulation of the potassium/chloride extruder KCC2. Plantier et al. [54] reported that calpain fosters the upregulation of the sustained current (INaP) in motor neurons and downregulation of KCC2 expression after SCI, resulting in the hyperexcitability of motor neurons after SCI, leading to spasticity. These findings support the possibility of using SBDPS as diagnostic and prognostic markers in SCI. 

The results of the present study also show that serum SBDP 150 and CSF SBDP 150/145 concentrations are significantly higher in patients with SCI. We could not measure the SBDP 145 levels in the serum due to the absence of an SBDP 145 human kit on the market. In patients with AIS grade A, higher serum SDP150 and CSF SBDP 150/145 levels were observed, although only CSF SBDP 150/145 concentrations measured with immunoblotting were found to be statistically significant. These results may indicate that different assay techniques or unmeasured SBDP 145 levels in the serum in patients with SCI may be important and should be acknowledged. Mondello et al. [34], in human subjects with TBI, found significantly higher CSF levels of SBDP 145 in those patients who died than in those who survived the injury [34]. To our knowledge, there is only one published paper that evaluated SBDPs as biomarkers in seven patients with acute traumatic SCI [16]. In that study, Yokobori et al. [16] reported that SBDP 150 levels were elevated in the CSF and serum, as measured with the south-western enzyme-linked immunosorbent assay. Further studies should be done on serum SBDP 145 levels in patients with SCI to clarify the potential as a diagnostic indicator of injury severity.

UCH-L1 is a small 27 kDa deubiquitinating enzyme involved in regulating brain protein metabolism and primarily exists in the neuronal cell body cytoplasm of neuronal cells [25,35,36]. Recently, poly-catecholamine nanofilms were used for the ultrasensitive immunosensing of UCHL-1 in biofluids in patients with SCI by Khetani et al. [36]. In a study conducted by Stukas et al. [37], CSF UCH-L1 levels were shown to reflect the SCI severity in a time-dependent manner and to predict the outcome. In that study, no differences in serum UCH-L1 levels were reported, and concern regarding the assay’s sensitivity was acknowledged. The CSF UCH-L1 levels significantly correlated with the diagnosis of SCI and reflected the injury severity in a time-dependent way. However, this assessment failed to show such a significant difference in both serum and CSF UCH-L1 levels in terms of predicting the neurological outcome.

S100B, an astroglial 11 kDa calcium-binding protein mainly produced by astrocytes and increased expression, signals an astrocytic reaction to neural injury. Our study found both CSF and serum S100B levels to peak on day 1, and there was a strong correlation with the diagnosis of SCI (Appendix A). Additionally, there was a statistically significant increase in CSF levels on days 2 and 3 in patients with initial AIS A as compared to those with initial AIS B. Lee et al. [43] showed that the serum S100B levels were higher in the SCI group (*n* = 20) than in the non-SCI group (*n* = 10). In their study, routine CT radiography revealed no definite abnormal findings in 50% of the patients with SCI who were evaluated in the emergency department. In another study conducted in 60 patients, Du et al. [42] reported that the serum levels of S100B increased and reached a peak on the fourth day and then declined gradually. They also reported that S100B has predictive value for neurological recovery after 6 months. Pouw et al. [48] reported that the mean S-100β concentration in motor complete paralysis patients (AIS A, B) was significantly higher compared with motor incomplete paralysis (AIS C, D) patients, but no clear differences in S100B concentrations between the different AIS grades were founded. Kwon et al. [41] found that CSF S100B levels distinguished significantly between different AIS grades of SCI and predicted long-term neurological outcomes in SCI patients as well. Therefore, not only do both of these astroglial markers show a rise over the acute phase of SCI, they can also be used to prognosticate the severity of the injury. It is known that S100B also exists in adipocytes and chondrocytes, resulting in limitations due to the non-specificity of S100B for the diagnosis of SCI [12], since SCI patients can sustain injuries to other organ systems.

CSF Tau concentrations are accepted to be a potential diagnostic and prognostic biomarker for SCI [18,41,44,45,48]. To our knowledge, no study has assessed serum Tau concentrations in patients with SCI. The present study showed that both the Tau serum and CSF concentrations significantly increased in the SCI group when compared with the healthy control group. In our study, the Tau CSF levels were significantly higher on day 5 in patients with initial AIS A as compared to those with initial AIS B. These findings were similar to a previous study in which higher levels of CSF Tau correlated with higher injury severity [41]. A study by Pouw et al. [48] on the other hand reported no significant correlations between CSF Tau levels and AIS grades in SCI patients. Kwon et al. [41] reported that the levels of CSF Tau protein were significantly different between those who improved in AIS grade over 6 months and those who did not improve. In our study, we did not find significant differences in CSF samples in those subjects that did not improve when compared with the AIS grade group that did improve.

GFAP is another biomarker, which is a skeletal intermediate filament protein found in astrocytes. It reflects the pathological state of astrogliosis and astrocytic injury [18,25,46]. GFAP is widely accepted as a diagnostic TBI biomarker [47]. A limited number of studies have shown elevated GFAP levels in the CSF and serum in patients with SCI [24,41,46,48,49].

CSF GFAP, which is the most studied biomarker in SCI, reflects injury to astrocytes. During immunoblotting, we found that its breakdown products in the CSF were the dominant species over the intact GFAP after SCI, mirroring a previous pilot study by Yokobori et al. [16]. Additionally, the CSF GFAP levels were significantly higher in patients with a more severe SCI at presentation (AIS A) on days 2, 3 and 4 (Figure 5). Previously, Kwon et al. [45] reported that CSF GFAP detected by ELISA at 24 h postinjury could predict, with 89% accuracy, future AIS-graded injury severities as well as 6 month postinjury segmental motor improvements in SCI patients. Kwon et al. [41] also showed that the CSF GFAP levels additionally served as predictors of future neurological outcomes in 50 acute SCI patients. In a study conducted by Pouw et al. [48], the CSF GFAP levels were measured within 24 h of injury in 16 SCI patients and were found to be significantly elevated in SCI patients when compared to healthy controls. Ahadi et al. [49] reported the mean serum levels of GFAP in SCI patients at 24, 48, and 72 h postinjury to be significantly higher than the healthy control levels. Furthermore, the serum GFAP levels and their analysis correlated with injury severity. Guéz et al. [46], on the contrary, did not find a significant elevation in CSF GFAP concentrations in SCI patients when compared to healthy controls. Our results agree with those of a recently published study by Stukas et al. [24], who reported in 113 acute SCI patients that the CSF GFAP concentrations were significantly elevated in SCI AIS grade A and B patients compared to controls.

NF-L, which is present in the axonal cytoplasm, reflects axonal injury, and its appearance is often used to evaluate neurological disorders and may be considered as a potential biomarker for SCI [24,25,46,50]. In the present study, the NF-L levels in both the CSF and serum increased and peaked on day 5 in SCI patients. In contrast, the Tau levels in the CSF were shown to return to baseline by day 5, and the levels of NF-L remained significantly elevated on day 5. Our results are similar to those of Kuhle et al. [50], who reported that the serum NF-L concentrations increased up to 7 d postinjury in SCI patients compared to healthy controls. Our results did not show statistically significant differences in NF-L levels between patients classified based on injury severity according to AIS grade A and AIS grade B. These findings are in contrast with Stukas et al.’s [24] study, which reported that the serum and CSF NF-L levels were increased with injury severity and were different between baseline AIS grade A and AIS grade B patients. Recently, Guéz et al. [46] reported that SCI patients with complete as compared to incomplete motor loss showed higher levels of CSF NF-L. The inability of our analysis to identify a correlation of NF-L levels with injury severity can be explained by the sample characteristic of our SCI patients, which all had a motor block.

Interleukin (IL)-6, an inflammatory cytokine, has also been reported to correlate with injury severity and outcomes following SCI [18,41]. In the present study, the serum and CSF IL-6 levels were significantly elevated when compared with the healthy control group. However, the levels of CSF IL-6 did not correlate with the injury severity in SCI patients. Our results are in agreement with Kosa et al. [51], who reviewed 13 published studies including 532 SCI and 437 healthy controls on inflammation-related markers. They found that the IL-6 levels were significantly higher in the SCI group than in the healthy control group [51]. Yang et al. [23] indicated that SCI injury did not affect IL-6 levels in either the CSF or serum in a rat model. In contrast, Dalkilic et al. [18] reported that the CSF IL-6 levels at 24 h postinjury correlated with injury severity and predicted the prognosis in SCI patients. Similarly, Kwon et al. [41] reported that the levels of CSF IL-6 at 24 h amongst AIS A patients postinjury were significantly different in those who did and did not improve over time.

In our study, no CSF biomarker showed a statistically significant difference between SCI patients with an improved AIS grade at discharge and those with no improvement. This result may be explained by the small study population as well as the low incidence of patients with SCI who required a lumbar catheter during enrollment, which occurred over the course of 4 years in a single medical center. Although the results of the current study showed strong correlations between biomarkers and the diagnosis of SCI with AIS grade A and B patients, we did not investigate AIS grade C and D patients. Thus, further validation in a larger cohort from multiple centers is warranted. Another limitation of the study was that all time points for sample collection were in the acute phase, while the chronic biomarkers for the prognosis of SCI were not explored. Additionally, SCI patients in the study were only scored using their AIS grades at admission and discharge, and there was no long-term follow-up after discharge.

We were also not able to assay the αII- spectrin, SBDP145, and GBDPs concentrations in the serum due to the absence of an available human kit for the ELISA. However, the CSF concentrations of αII-spectrin (day 1, *p* > 0.05), SBDP145/150 (day 1 to day 4, *p* < 0.05), and GBDP (day 1 and day 2, *p* < 0.005) were significantly higher when compared with the control group. Future studies should be conducted to clarify those biomarker serum levels after SCI. Lastly, we used different detection methods, such as immunoblotting, ELISA, and digital SIMOA N4PB ELISA, to measure the levels of biofluid biomarkers. We noted that when the GFAP values were measured with immunoblotting, no significant differences between the AIS grade A and AIS grade B patients could be demonstrated. However, there was a significant difference when the measurement was performed with the ELISA. These divergent results imply that more sensitive measurements need to be used in the future to provide the required sensitivity for assessing future targets for drug discovery. In addition, a standardization strategy is needed when comparing the results of studies across different assay methodologies. Finally, as commonly emphasized in clinical SCI studies, future prospective controlled studies including large sample sizes and long-term outcomes should be conducted to help advance the important field of SCI diagnosis and treatment. Despite these limitations, this study demonstrates a strong correlation between biofluid biomarkers and the diagnosis of SCI. Future prospective controlled studies including large sample size and long-term outcomes should be done to support the results of this study.

## Figures and Tables

**Figure 1 diagnostics-13-01814-f001:**
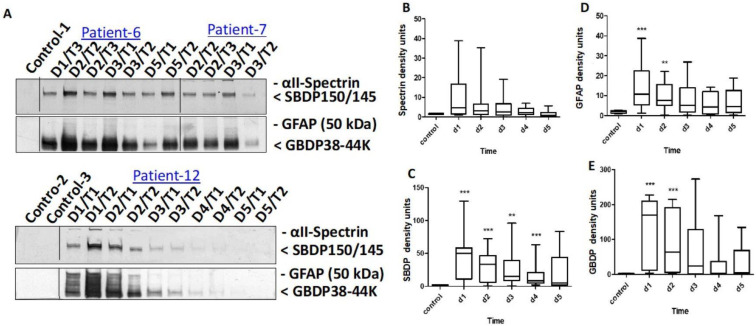
CSF biomarker (αII-spectrin, SBDP150/145, GFAP, GBDP 38/44K) levels measured using immunoblotting in the diagnosis of SCI and comparison between SCI and healthy control groups: (**A**) representative images from immunoblotting for αII-spectrin, SBDP 150/145, GFAP, and GBDP38/44K (**B**–**E**); quantification shown as the median, interquartile range (box), and upper and lower values (whiskers) for αII-spectrin (**B**), SBDP150/145 K (**C**), GFAP (**D**), and GBDP 38/44K (**E**). ** Denotes statistical results compared between patients with SCI (*n* = 15) and healthy control (*n* = 10) groups. Note: **, *** *p* values < 0.05 and 0.005 vs. control.

**Figure 2 diagnostics-13-01814-f002:**
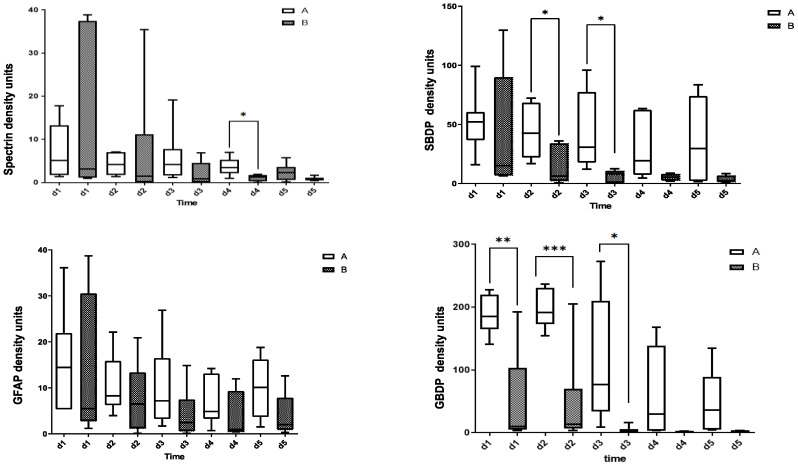
Ability of CSF αII-spectrin, SBDP150/145, GFAP, and GBDP 38/44K levels to predict injury severity (AIS grade) measured by immunoblotting. * Denotes statistical results compared to initial AIS grade A (*n* = 11) and AIS grade B (*n* = 6) patients in SCI group. Quantification is shown as the median, interquartile range (box), and upper and lower values (whiskers). Note: *, **, *** *p* values < 0.05, 0.01 and 0.001 between groups.

**Figure 3 diagnostics-13-01814-f003:**
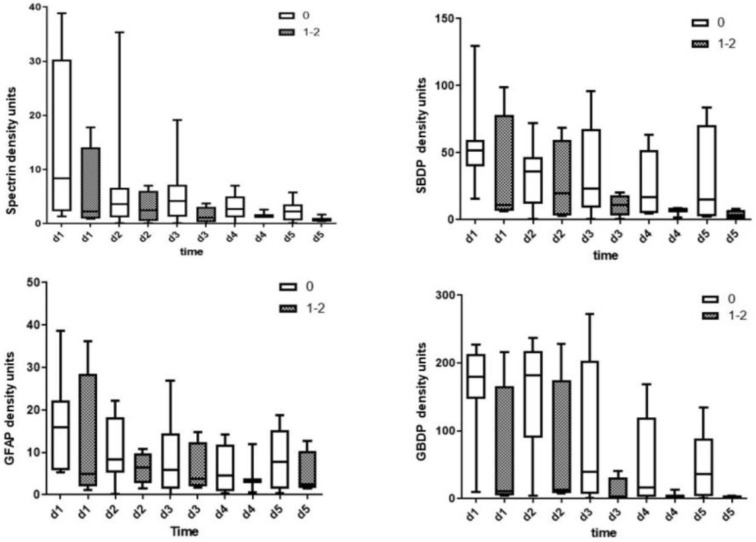
The ability of CSF biomarker levels measured with immunoblotting to predict AIS improvements (ΔAIS = discharge AIS − initial AIS; ΔAIS 0 = no changes; ΔAIS 1–2 = improvement in AIS grade 1 or 2.

**Figure 4 diagnostics-13-01814-f004:**
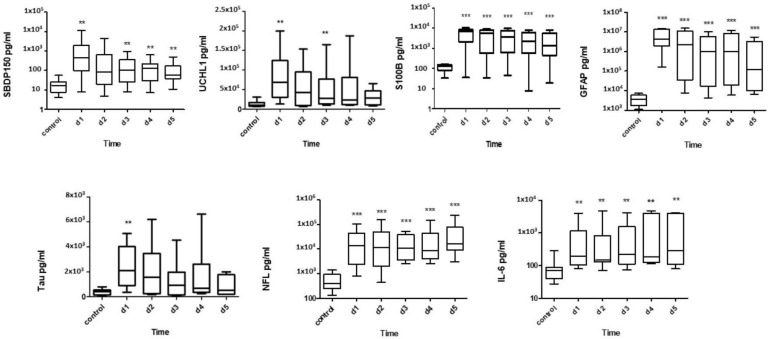
CSF biomarker (SBDP150, UCH-L, S100B, Tau, GFAP, NF-L, IL-6) levels measured by ELISA in the diagnosis of SCI and a comparison between SCI and healthy control groups. ** Denotes statistical results compared between patients with SCI (*n* = 15) and healthy control (*n* = 10) groups. Quantification is shown as median, interquartile range (box), and upper and lower values (whiskers). Note: **, *** *p* values < 0.05, 0.005 vs. control.

**Figure 5 diagnostics-13-01814-f005:**
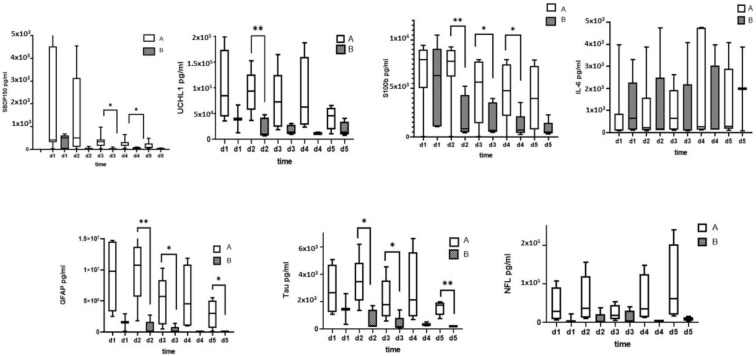
Ability of CSF SBDP150, UCH-L, S100β, Tau, GFAP, NF-L, and IL-6 levels to predict injury severity (AIS grade) measured by ELISA. * Denotes statistical results compared to initial AIS grade A (*n* = 11) and AIS grade B (*n* = 6) patients in SCI group. Quantification is shown as the median, interquartile range (box), and upper and lower values (whiskers). Note: *, ** *p* values < 0.05, and 0.01 between AIS grade A and grade B patients.

**Figure 6 diagnostics-13-01814-f006:**
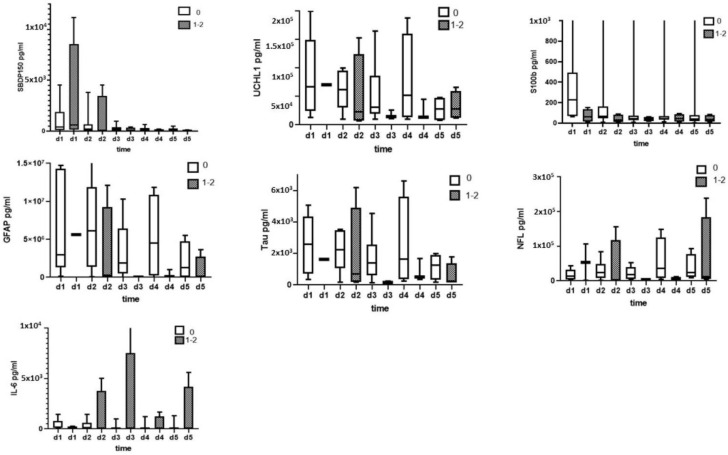
The ability of CSF biomarker (CSF SBDP150, UCH-L, S100B, Tau, GFAP, NF-L, and IL-6) levels measured by ELISA to predict AIS improvements (ΔAIS = discharge AIS − initial AIS; ΔAIS 0 = no changes, ΔAIS 1–2 = improvement in AIS grade 1 or 2 grade).

**Figure 7 diagnostics-13-01814-f007:**
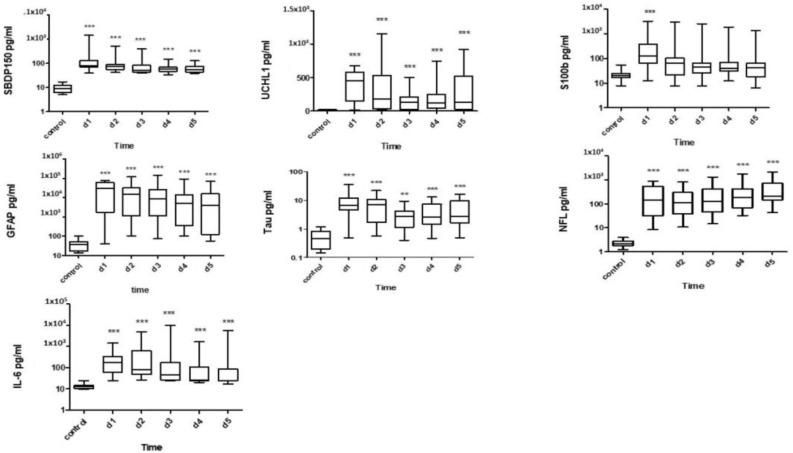
** Denotes statistical results compared between patient with SCI (*n* = 15) and healthy control (*n* = 10) groups. Quantification shown as median, interquartile range (box), and upper and lower values (whiskers). Note: **, *** *p* values < 0.05, 0.005 vs. control.

**Figure 8 diagnostics-13-01814-f008:**
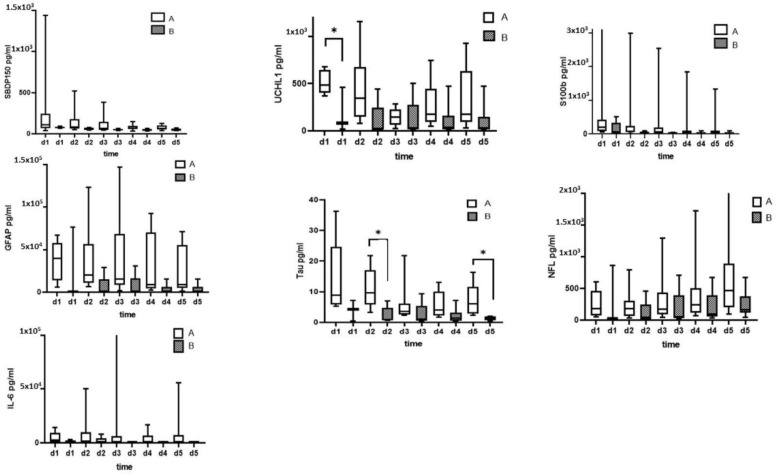
The ability of serum SBDP150, UCH-L, S100B, Tau, GFAP, NF-L, and IL-6 levels to predict injury severity (AIS grade) was measured by ELISA. * Denotes statistical results compared to initial AIS grade A (*n* = 11) and AIS grade B (*n* = 6) patients in the SCI group. Quantification is shown as the median, interquartile range (box), and upper and lower values. Note: * *p* < 0.05.

**Figure 9 diagnostics-13-01814-f009:**
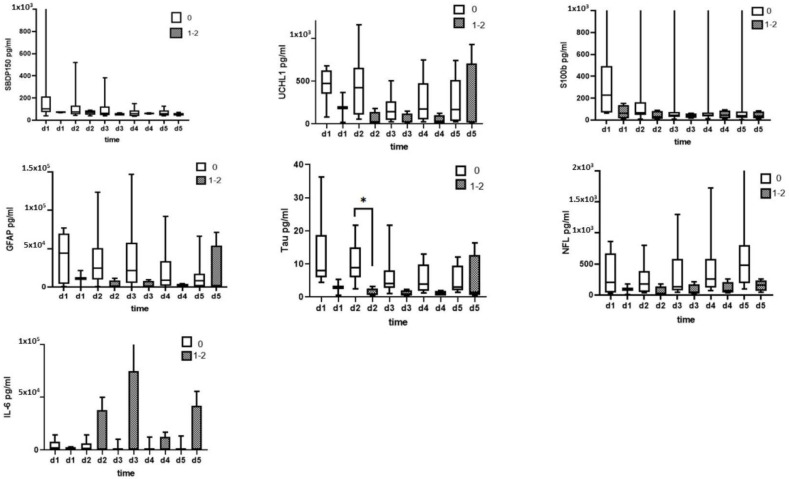
Ability of serum biomarker (SBDP150, UCH-L, S100B, Tau, GFAP, NF-L, and IL-6) levels measured by ELISA to predict AIS improvement (ΔAIS = initial AIS − discharge AIS; ΔAIS 0 = no changes; ΔAIS 1–2 = improvement in AIS grade 1 or 2 grade). Note: * *p* < 0.05.

**Table 1 diagnostics-13-01814-t001:** Characteristics of patients with spinal cord injury. AIS = American Spinal Injury Association impairment grade I-AIS = initial AIS grade; D-AIS = discharge AIS grade; A = motor and sensory complete paralysis; B = motor complete, sensory incomplete paralysis; C = incomplete motor and sensory paralysis; D = incomplete motor paralysis; MVA = motor vehicle accident; C = cervical; T = thoracic.

Patient	Age	Gender	Cause and Level of Injury	I-AIS	D-AIS	ΔAIS (I-AIS-D-AIS)
P1	43	Male	Pedestrian MVA, C7-T1	A	A	0
P2	38	Male	Fall, T3-T4	B	D	2
P3	21	Male	Diving accident C5-C6	B	D	2
P4	28	Male	Diving accident C5-C6	B	B	0
P5	19	Male	MVA, C5-C6	B	C	1
P6	22	Male	Sport Injury C5	A	A	0
P7	48	Male	MVA, T3	A	A	0
P8	38	Female	MVA, T1/2	A	A	0
P9	29	Female	MVA, T12	A	A	0
P10	57	Male	Fall, T11-12	A	A	0
P11	67	Male	Fall, C3-C4	A	A	0
P12	22	Male	Sport Injury C5-C6	A	B	1
P13	66	Male	Fall T12-L1	B	B	0
P14	22	Male	Gunshot T-12	A	A	0
P15	58	Male	Fall, T4-T5	B	B	0

## Data Availability

The data presented in this study are available on request from the corresponding author.

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
