# Peer review of "Association between Cerebrospinal Fluid and Serum Biomarker Levels and Diagnosis, Injury Severity, and Short-Term Outcomes in Patients with Acute Traumatic Spinal Cord Injury"

_diagnostics, 2023, doi:10.3390/diagnostics13101814_

Round 1

Reviewer 1 Report

The aim of the this prospective controlled study was to determine the CSF and serum profile of 10 biomarkers as diagnosis, severity and prognosis of SCI. 
The main concern is the small sample size and the   Weakness of introduction to explain the rationale of study. There are several studies in this field with a larger sample size. 
All abbreviations should be explained for the first use.

Author Response

Please see the attached response.

Reviewer 2 Report

Thank you for the opportunity to review this manuscript. In this manuscript authors have presented analyses of CSF biomarkers in patients with acute traumatic spinal cord injury. There are certain issues that need to be addressed before the manuscript can be considered for publication. Firstly, I would recommend that authors make corrections to their analysis for multiple comparisons. Given the small sample size, there are chances of type I error. Correction for multiple comparisons will make analysis more robust. Secondly, authors have provided IRB number as “….”, which needs to be corrected. Lastly, there are grammatical errors and spelling mistakes and frequent fonts changes at certain places without any obvious need. These need to be corrected before the manuscript can be considered for publication.

Author Response

Please see the attached response.

Round 2

Reviewer 1 Report

All of my comments have responded satisfactorily. 

Author Response

Please see the attached revised version. Thank you.
